# Radiomics-Based Classification of Tumor and Healthy Liver on Computed Tomography Images

**DOI:** 10.3390/cancers16061158

**Published:** 2024-03-14

**Authors:** Vincent-Béni Sèna Zossou, Freddy Houéhanou Rodrigue Gnangnon, Olivier Biaou, Florent de Vathaire, Rodrigue S. Allodji, Eugène C. Ezin

**Affiliations:** 1Université Paris-Saclay, UVSQ, Univ. Paris-Sud, CESP, Équipe Radiation Epidemiology, 94805 Villejuif, France; florent.devathaire@gustaveroussy.fr (F.d.V.); rodrigue.allodji@gustaveroussy.fr (R.S.A.); 2Department of Clinical Research, Radiation Epidemiology Team, Gustave Roussy, 94805 Villejuif, France; 3Centre de Recherche en Épidémiologie et Santé des Populations (CESP), U1018, Institut National de la Santé et de la Recherche Médicale, 94805 Villejuif, France; 4Ecole Doctorale Sciences de l’Ingénieur, Université d’Abomey-Calavi, Abomey-Calavi 384, Benin; 5Department of Visceral Surgery, CNHU-HKM, Cotonou 229, Benin; fredgnang@yahoo.fr; 6Department of Radiology, CNHU-HKM, Cotonou 229, Benin; biaouolivier@gmail.com; 7Institut de Formation et de Recherche en Informatique, Université d’Abomey-Calavi, Abomey-Calavi 384, Benin; eugene.ezin@uac.bj; 8Institut de Mathématiques et de Sciences Physiques, Université d’Abomey-Calavi, Dangbo 384, Benin

**Keywords:** radiomic features, classification, machine learning, liver lesions, hepatocellular carcinoma, metastasis

## Abstract

**Simple Summary:**

Liver malignancies, particularly hepatocellular carcinoma, and metastases stand as prominent contributors to cancer mortality. Within abdominal computed tomography imaging, much of the data remain underused by radiologists. Radiomics uses advanced image analysis to extract quantitative features from medical scans for deeper diagnosis, treatment, and prognosis insights. Machine learning algorithms enable analyzing these features, facilitating an automatic, rapid, and efficient medical management process. We used these algorithms to train models that can distinguish between healthy livers and those with tumors, as well as between HCC and metastatic tumors, using CT images from the electronic medical records of the Centre National Hospitalier Universitaire Hubert Koutoukou Maga (CNHU-HKM) in Benin. The high correlation scores suggest that the radiomics signature is a prognostic biomarker for hepatic tumor screening.

**Abstract:**

Liver malignancies, particularly hepatocellular carcinoma and metastasis, stand as prominent contributors to cancer mortality. Much of the data from abdominal computed tomography images remain underused by radiologists. This study explores the application of machine learning in differentiating tumor tissue from healthy liver tissue using radiomics features. Preoperative contrast-enhanced images of 94 patients were used. A total of 1686 features classified as first-order, second-order, higher-order, and shape statistics were extracted from the regions of interest of each patient’s imaging data. Then, the variance threshold, the selection of statistically significant variables using the Student’s *t*-test, and lasso regression were used for feature selection. Six classifiers were used to identify tumor and non-tumor liver tissue, including random forest, support vector machines, naive Bayes, adaptive boosting, extreme gradient boosting, and logistic regression. Grid search was used as a hyperparameter tuning technique, and a 10-fold cross-validation procedure was applied. The area under the receiver operating curve (AUROC) assessed the performance. The AUROC scores varied from 
0.5929
 to 
0.9268
, with naive Bayes achieving the best score. The radiomics features extracted were classified with a good score, and the radiomics signature enabled a prognostic biomarker for hepatic tumor screening.

## 1. Introduction

Liver cancer is the sixth-most common and third-deadliest cancer worldwide, with projections suggesting an increase to over a million cases by 2030. It primarily affects males, especially in regions with high hepatitis B prevalence, such as Asia and Sub-Saharan Africa. Hepatocellular carcinoma (HCC) and intrahepatic cholangiocarcinoma make up 99% of liver cancer cases [1]. Early diagnosis, crucial for managing liver diseases, relies significantly on diagnostic imaging to detect primary tumors or metastases and monitor tumor progression.

High-resolution imaging modalities such as magnetic resonance imaging (MRI), ultrasound, and computed tomography (CT) help detect liver tumors. Radiologists and oncologists most frequently use computed tomography (CT) for liver lesion evaluation and staging [2]. Radiomics is a non-invasive method for determining the relationship between quantitative medical picture characteristics and underlying biological events [3]. The use of radiomics-based computer-aided diagnosis (RCAD) [4] from CT images significantly enhances diagnostic processes by providing an objective and rapid assessment, which is particularly beneficial for less experienced or less confident radiologists. Additionally, it is especially relevant in settings where CT scans are a primary tool for routine checkups, making it a valuable asset for medical facilities with varying resources.

The approach to liver cancer treatment varies based on the tumor’s phenotype [5,6]. Initially, radiologists employed MRI or CT scans to assess the phenotype, and various manual grading criteria for liver tumors have been established [7]. However, the visual characteristics of these lesions on scans can greatly differ due to histological variations [8], leading to subjective interpretations and a lack of consensus on their exact definitions [7]. While tissue biopsy offers definitive histological confirmation, it is not routinely required for HCC diagnosis thanks to advancements in non-invasive methods utilizing characteristic imaging features and radiomics [9].

Many RCAD studies have been conducted and published. In 2007, S. G. Mougiakakou and others [10] used the gray-level histogram (GLH), the gray-level co-occurrence matrix (GLCM), and other methods to extract texture features from CT images. They then built models using support vector machine (SVM), *K*-nearest neighbors, the probabilistic neural network (PNN), and other methods to help with the diagnosis of normal liver, hepatic cyst, hepatic hemangioma (HEM), and hepatocellular carcinoma (HCC). In 2013, S. S. Kumar et al. [11] used the GLH, GLCM, wavelet transform, contourlet transform, and their fusions to extract texture data from CT images and then constructed PNN models for classifying HCC and HEM. J. J. Qiu et al. [12] investigated the same classification task, applying SVM models to data extracted from plain CT images with the methods of the GLH, GLCM, etc. In 2017, Y. Zhou et al. [13] used the Laplacian of Gaussian (LoG) spatial band-pass filter, GLH, and GLCM to extract textural data from arterial- and portal venous-phase CT images and, then, constructed logistic regression (LR) models to identify early recurrence of HCC within a year. In 2020 and 2021, Nie et al. [14,15] designed a nomogram based on CT radiomics to differentiate HCC from focal nodular hyperplasia (FNH) and hepatocellular adenoma (HA) in the normal non-cirrhotic liver with high accuracy. In 2022, S. Lysdahlgaard used morphological and statistical methods along with LR, random forest (RF), and SVM models to obtain radiomic features like the size, shape, and location of the tumor and healthy liver tissue [16].

Recent research has demonstrated significant potential in using radiomics to identify variables associated with clinical outcomes [17,18]. However, only a few studies have used the entire liver as the region of interest (ROI) in their research. Additionally, no African investigations have been conducted on applying radiomics or machine learning to classify liver lesions. This study evaluates the effectiveness of radiomic features extracted from CT scans of the liver in distinguishing between healthy and cancerous tissues and differentiating HCC from metastatic tumors, thereby leveraging radiomics for nuanced diagnostic insights into liver health. To our knowledge, this is the first study in Africa.

## 2. Materials and Methods

### 2.1. Proposed System

A machine learning (ML) model has been developed to accurately classify healthy livers and livers with tumors using radiomic features extracted from CT images. There are five stages in the workflow process of the proposed system: segmentation of ROIs, feature extraction, statistical analysis, feature selection, and classification. The entire liver was interactively evaluated as the ROI. Figure 1 shows the block diagram of the proposed system. The work consists of two main phases: classifying healthy and tumor livers. Then, among those with tumors, classifying them as HCC or metastases.

### 2.2. Dataset Description

In this study, 94 CT images obtained from the electronic medical record of the Centre National Hospitalier Universitaire Hubert Koutoukou Maga (CNHU-HKM) for January 2018 to December 2021 were retrospectively included. The dataset reveals a diverse array of tumor sizes, with dimensions ranging from a minimum of 
3.082
 mm to a maximum of 
144.4
 mm and a median value of 
11.18
 mm. The cohort comprises 38 healthy patients, 23 confirmed to have HCC and 33 having metastatic tumors. Each scanner folder is an abdominal CT scan including four injection phases of contrast enhancement: 15–20 s for the early arterial phase, 45 s for the late arterial phase, 80 s for the portal venous phase, and 4 min for the equilibrium delay phase. Only the portal phase was used because it can show complex structural details, improve vascular clarity, boost tissue contrast, and make it easier to find lesions. The imaging dataset’s details are reported in Table 1.

### 2.3. ROI Segmentation

Image segmentation is the procedure by which an image is partitioned into significant regions, with distinct attributes assigned to individual pixels. The performance of image-segmentation methods is contingent upon various factors, including the type of image, manner of application, size, and color intensity. Two radiologists, each with three years of experience in abdominal CT diagnosis, used the Radiology Informatics Laboratory Contour (*RIL-Contour*) software [19] to segment the ROI in the images manually. Figure 2 shows a sample CT scan of a liver tumor and the segmentation.

### 2.4. Intensity Normalization

Variations in CT image intensity can occur, even when the same CT scanner and identical scanning parameters are used. These variations may affect the process of extracting features from the images. Furthermore, the primary image may be blurred, leading to potential confusion between the ROI and other organs or regions present in the image. The dataset was preprocessed to improve the contrast of the images. The Hounsfield unit (HU) is a relative quantitative measurement of radiodensity (a property of a tissue to attenuate/absorb an X-ray beam) that radiologists use to interpret CT images. The HU values typically vary between 
−1000
 and 3000 [20]. The HU values of the image were set to the range 
[150,300]
 in this study.

### 2.5. Feature Extraction

The main goal of radiomics is to create a machine learning algorithm that can categorize outcomes based on objective criteria, utilizing quantitative data from medical images. This work comprehensively analyzed 1686 radiomics features per patient, classified into four categories: first-order, second-order, higher-order, and shape features (Table 3). First-order features can accurately depict the spatial distribution of various voxel intensities, irrespective of the three-dimensional configuration [21]. The attributes of the second-order statistics can be identified by analyzing a density histogram as follows:(i)The gray-level co-occurrence matrix (GLCM) that quantifies an image’s texture by computing the frequency of pairs of pixels with given values and a defined spatial connection [22].(ii)The gray-level run length matrix (GLRLM), a matrix that can extract texture features for texture analysis. The GLRLM technique is utilized to extract advanced statistical texture properties [23].(iii)The gray-level size zone matrix (GLSZM) measures the number of gray-level zones in an image. A gray-level zone is a group of connected voxels with the same gray-level intensity [24].(iv)The gray-level dependence matrix (GLDM), which measures the relationships between gray levels in an image. There is a gray-level dependency when a count of connected voxels within a distance 
δ
 of the center voxel depends on it [25].

Based on the first-order and second-order features, filter grids were applied to the images, and transformed features, namely higher-order statistical features, were obtained. Six filters were applied: exponential, square, square root, logarithm, Laplacian of Gaussian (LoG), and wavelet. Each filter was selected for its capacity to enhance or reveal specific aspects of imaging data, contributing to a more comprehensive and precise analysis. The exponential filter intensifies low-amplitude variations in the image, enabling the better distinction of subtle regions that might be significant for radiomic analysis. The square filter amplifies high intensities, while the square root filter mitigates high intensities, making low-contrast areas more discernible. The logarithmic filter compresses the dynamic range of high-dynamic images, facilitating the visualization and analysis of fine details in bright and dark regions. The LoG is particularly effective for edge detection and contour enhancement. It accentuates regions with rapid changes in intensity, which is crucial for identifying lesion boundaries or other relevant structures. The sigma values for LoG were 
0.5
, 
1.0
, 
2.0
, 
3.0
, 
4.0
, and 
5.0
. Wavelet analysis decomposes the image into components at different scales, allowing for extracting pertinent information that other methods might miss. This is particularly useful for capturing localized and multi-scale features in imaging data. The wavelet transforms and decomposes an image into a series of sub-images, each representing different frequency components (low (L) and high (H)) along different orientations (horizontal, vertical, and diagonal). L denotes low frequency (smooth parts of the image), and H denotes high frequency (edges and fine details). Based on these frequencies, eight levels of wavelet decomposition have been obtained: wavelet-LHL, wavelet-LHH, wavelet-HLL, wavelet-LLH, wavelet-HLH, wavelet-HHH, wavelet-HHL, and wavelet-LLL). The extracted features are shown in Table 2.

### 2.6. Statistical Analysis and Feature Selection

Three steps were followed to select the most relevant features. In the first step, a variance criterion was set to decrease the number of features that did not meet the consistency threshold, which was set to 
0.8
 [21]. Variance thresholds remove features whose values do not change much from observation to observation (i.e., their variance falls below a threshold). These features provide little value. In the second step, based on the Student’s *t*-test, the features that did not show statistical differences (
p>0.05
) were removed. The *t*-test is any statistical hypothesis test in which the test statistic follows a Student’s *t*-distribution under the null hypothesis [26]. This process can be viewed as preconditioning the predictive model [21]. The last step was penalization with the lasso algorithm [27]. The best value of lambda was identified, and the variance inflation factor (VIF) was calculated to obtain the most relevant features. The VIF measures the severity of multicollinearity in multiple regression models. It represents the ratio of the estimator variance of the regression coefficient to the variance when no linear correlation between the independent variables is assumed. The VIF can be calculated as follows:
(1)
$$VIF=\frac{1}{1-R_{i}^{2}}$$

where 
$R_{i}$
 is the negative correlation coefficient of the regression analysis for other independent variables. The larger the VIF, the greater the possibility of collinearity between the independent variables. Generally, multicollinearity is assumed when the VIF value is greater than five. Thus, removing the radiomic features with a VIF value greater than five is necessary. Statistical outcomes for the variable of age were derived from a *t*-test analysis, while the results for the variable of sex were obtained using a *chi*-*squared* test. To build the classification model, the segmentation was labeled as two categories: tumor tissue and normal tissue. All steps were implemented using the *R* software version 
4.3.2.


### 2.7. Models and Evaluation

Six supervised machine learning techniques were used for the classification, logistic regression (LR), random forests (RF), support vector machine (SVM), extreme gradient boosting (XGBoost), adaptive boosting (AdaBoost), and naive Bayes. The 10-fold cross-validation approach was used with random data splitting (
70%
 for training and 
30%
 for test). For the hyper-parameters’ optimization, a grid search was used for each model [28]. The optimal parameters were found as follows:SVM: “*C*” = 0.05050505;Naive Bayes: “
laplace
” = 0, “
adjust
” = 1.5, “
usekernel
” = 
TRUE
;XGBoost: “
rounds
” = 100, “
max_depth
” = 3, “
eta
” = 0.01, “
gamma
” = 0.2, “
colsample_bytree
” = 0.5, “
min_child_weight
” = 3, “
subsample
” = 1;RF: “
mtry
” = 21;LR: “
alpha
” = 1, “
lambda
” = 
0.1
;AdaBoost: “
iter
” = 150, “
maxdepth
” = 5, “
nu
” = 1.

The models’ performances were evaluated using the area under the receiver operating curve (AUROC), sensitivity, specificity, positive predictive value (PPV), negative predictive value (NPV), and Matthews correlation coefficient (MCC). The best radiomics model was then screened. The AUROC predictive model’s performance is presented as 
95%
 corrected confidence intervals (CIs). The normal liver class was used as the positive class by the models.

## 3. Results

### 3.1. Clinical Factors of Patients

According to the patient’s clinical records, sex and age were used as clinical characteristics. Table 3 shows the statistical results between the tumor and non-tumor groups.

### 3.2. Radiomic Feature Selection

The dataset comprised diverse tumor contrast levels, variations in tissue size abnormalities, and various quantities of lesions. Validation was performed for the segmentations of each CT scan. There were 1686 features, including original, exponential, square, square root, logarithm, LoG, and wavelet transformations, extracted, as presented in Table 2. After calculations based on the variance threshold, 804 features underwent the first dimensionality reduction. Then, features with a *p*-value > 
0.05
 were excluded by computing the Student’s *t*-test, and 487 features were selected. Among them, 28 features with non-zero coefficients were retained after lasso logistic regression analysis. Their VIF was calculated, and the variables with a 
VIF>5
 were removed to avoid model over-fitting. A total of seven radiomics features were retained as a result. The VIFs were all smaller than 5, indicating no multicollinearity among the seven radiomic features. Table 4 presents these variables with their VIF values. Figure 3 compares the selected variables between liver and tumor tissue. A correlation heat map of the selected radiomic features is presented in Figure 4.

### 3.3. Construction and Evaluation of the Classification Model of Tumor Presence or Not

The naive Bayes model achieved the best performance among the other algorithms, with an AUROC of 0.9268 (95% CI: 0.8224–1), sensitivity of 
0.9091
, specificity of 
0.9444
, NPV of 
0.9444
, PPV of 
0.9091
, and MCC of 
0.8536
. AdaBoost and XGBoost performed similarly, with an AUROC of 0.8813 (95% CI: 0.75–1). SVM was the worst algorithm, with an AUROC of 0.8258 (95% CI: 0.677–0.9745). All scores obtained during the test phase are presented in Table 5. The ROC curves of all models are plotted in Figure 5. Twenty-two radiomic features showed a higher importance correlation and are presented in Table 6.

### 3.4. Construction and Evaluation of the Classification Model of HCC and Metastatic Tumors

The naive Bayes model achieved the best performance again, among the other algorithms, with an AUROC of 0.8571 (95% CI: 0.6764–1), sensitivity of 
0.7143
, specificity of 1, NPV of 
0.8333
, PPV of 1, and MCC of 
0.7715
. It was followed by the lasso regression with an AUROC of 0.7569 (95% CI: 0.5472–0.9667). All other models performed similarly, with an AUROC of 0.5929 (95% CI: 0.3873–0.7985). All scores obtained during the test phase are presented in Table 7. The ROC curves of all models are plotted in Figure 6. Eighteen radiomic features showed a higher importance correlation and are presented in Table 8.

## 4. Discussion

This study shows that machine learning algorithms can capture the difference between normal and tumor liver tissue and can detect HCC and metastatic tumors in CT images at the portal phase using radiomic features. In radiomics, a significant array of features, often numbering in the hundreds, is generated to delineate a specified ROI through various methodologies [29]. These features undergo evaluation for their potential as prognostic indicators. Additionally, the critical process of feature selection demands a focus on their consistency and responsiveness concerning the delineation methodology for them to be deemed clinically viable. To date, no studies have systematically assessed the application of various machine learning algorithms in differentiating between tumor and non-tumor tissues using the entire liver as the ROI. Furthermore, no recorded African studies have employed machine learning to classify liver lesions.

The approach used six machine learning algorithms, enhanced with hyper-parameter tuning through grid search and validated via 10-fold cross-validation to identify the most effective model based on the extracted features. This method ensures the selection of the best model by systematically exploring a range of parameter settings and rigorously evaluating model performance across multiple subsets of the data. The study revealed that the naive Bayes algorithm outperformed the others, achieving the highest marks across the AUROC, sensitivity, specificity, NPV, PPV, and MCC metrics for the two classification tasks. This demonstrates the reliability of the data and the stability of the naive Bayes model in this study. In detecting the presence of a tumor or not, AdaBoost and XGBoost were closely matched as the second-best performing algorithms, demonstrating equivalent outcomes in their performance evaluations. However, all models performed well, with no drastic difference in the scores.

In contrast, only naive Bayes and LASSO regression yielded satisfactory scores for detecting HCC and metastatic tumors. The other models produced low scores. In total, 1686 features were extracted from portal-phase CT images. Twenty-two features showed a higher correlation in the presence of tumor or no tumor classification, while there were eighteen in the classification of HCC and metastatic tumors.

Applying lasso regression for variable selection identified seven variables with a VIF below 5. The correlation map in Figure 4 demonstrates no correlation between features. The boxplot in Figure 3 demonstrates that most radiomic features extracted from tumor tissue cover a broader spectrum than those derived from normal liver tissue (5 features out of 7). The study of Lysdahlgaard observed the same phenomenon [16]. Zhou et al. [30] conducted a similar study, classifying gross tumor volume and normal liver tissue in HCC. They also identified seven variables with a VIF below 5. However, their findings differed in model performance; they identified naive Bayes as the least effective model, whereas XGBoost emerged as the superior algorithm in terms of AUC performance.

Age and sex were added to the selected features. It should be noted that there was no statistically significant difference in age and sex between healthy livers and those with tumors (
p>0.05
). However, age had a *p*-value of 
0.182
, suggesting that it may merit more examination in future studies. This study demonstrates the feasibility of using the entire liver as the region of interest for segmentation rather than extracting the local lesion for observations. It can be useful for automatically and swiftly identifying ill patients. It is particularly interesting because liver and lesion segmentation research yields more accurate results when segmenting the liver than the lesions themselves.

Nevertheless, it remains important to distinguish between lesions such as HCC, hepatic cyst, HEM, FNH, HA, etc., to administer appropriate treatments and avoid unnecessary procedures. Stratifying tumors for specific lesion types would have been ideal; however, the study’s small sample size limited this approach. Additionally, complementing hepatic CT with ultrasound or MRI could further enhance diagnostics, potentially unveiling new radiomic features from these modalities and improving lesion characterization and treatment accuracy. A study conducted by Yao et al. explored 2560 radiomic features from 177 patients’ multimodal ultrasound images and found that radiomic models could be useful for evaluating hepatic tumors, especially for diagnosis, differential diagnosis, and clinical prognosis [31]. By comparing the radiomic properties of ultrasound, MRI, and computed tomography, it may be possible to predict and differentiate between various types of hepatic lesions.

## 5. Conclusions

This research conducted a preliminary radiomics investigation using CT scans. A correlation was established between radiomic features and the distinction between healthy and tumor liver tissue, on the one hand, and between HCC and metastatic tumors, on the other hand. The naive Bayes model shows great results for the two tasks. It achieved an AUROC of 0.9268 (95%CI: 0.8224–1) in the distinction between healthy and tumor liver tissue and an AUROC of 0.8571 (95%CI: 0.6764–1) in the distinction between HCC and metastatic tumors. This result demonstrates that the developed radiomics signature is statistically significantly correlated with healthy liver tissue and liver tumor tissue using the entire liver as a ROI. However, there is a need for larger retrospective and prospective research on liver CT scans that examines potential prognostic markers and has strict reference criteria.

## Figures and Tables

**Figure 1 cancers-16-01158-f001:**
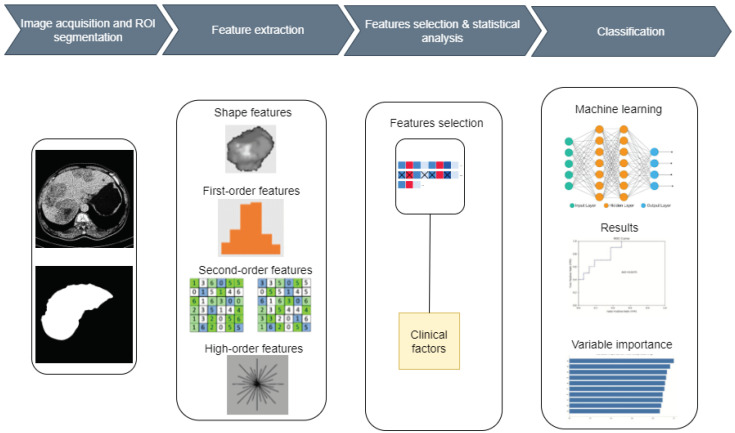
The block diagram of the proposed system.

**Figure 2 cancers-16-01158-f002:**
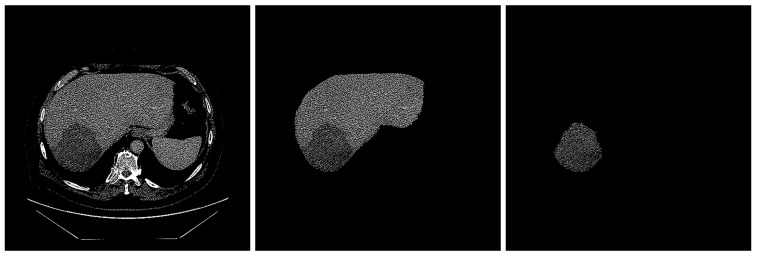
The segmentation of liver. (**left**) Original image with HCC masses inside. (**middle**) Liver segmented. (**right**) Tumor segmented.

**Figure 3 cancers-16-01158-f003:**
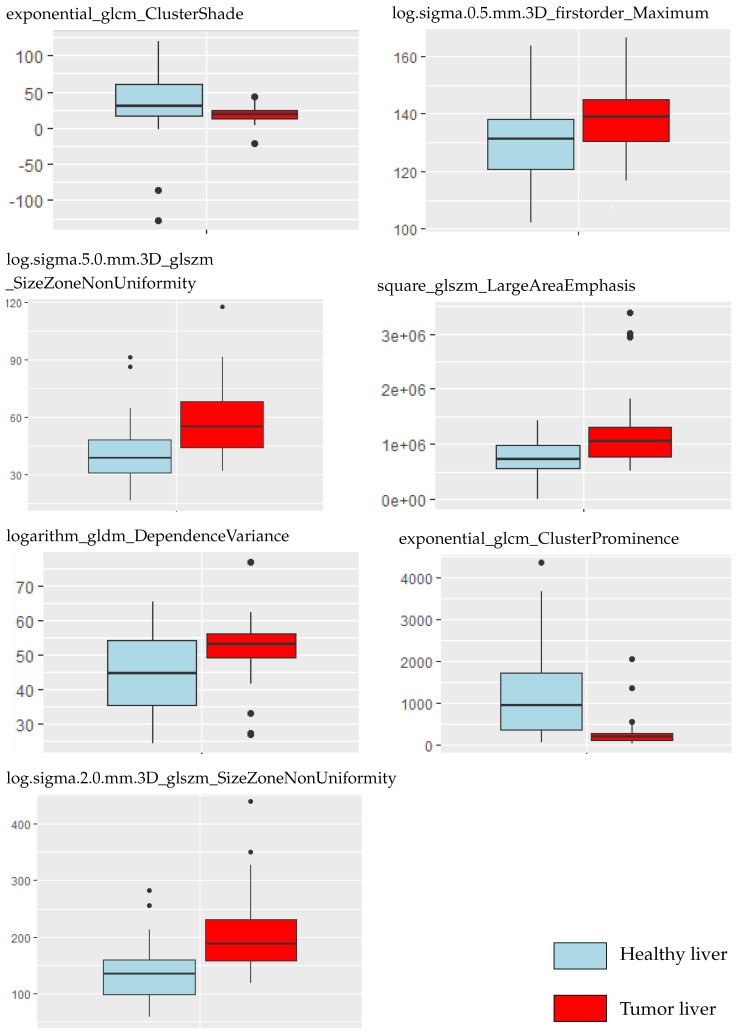
Selected radiomic features compared between tumor and non-tumor tissue.

**Figure 4 cancers-16-01158-f004:**
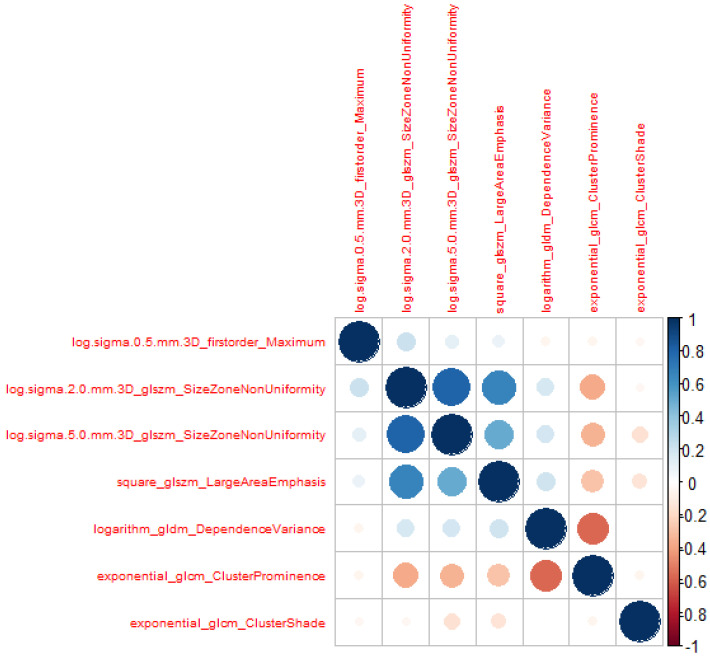
Heat map correlation of the selected features.

**Figure 5 cancers-16-01158-f005:**
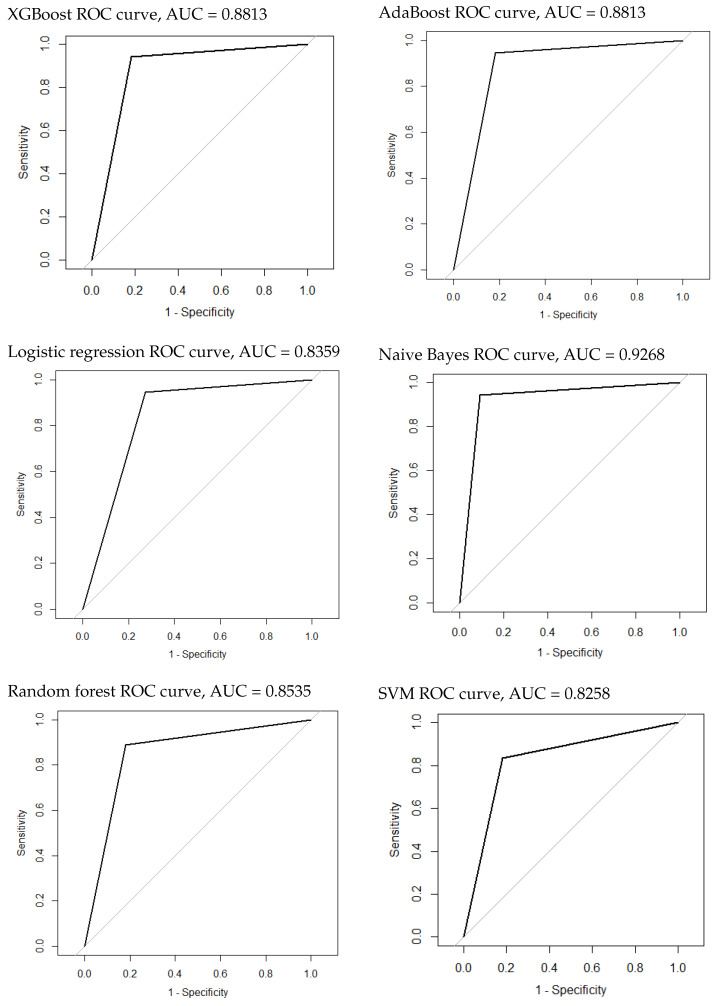
The ROC curves of the classification model of tumor presence or not.

**Figure 6 cancers-16-01158-f006:**
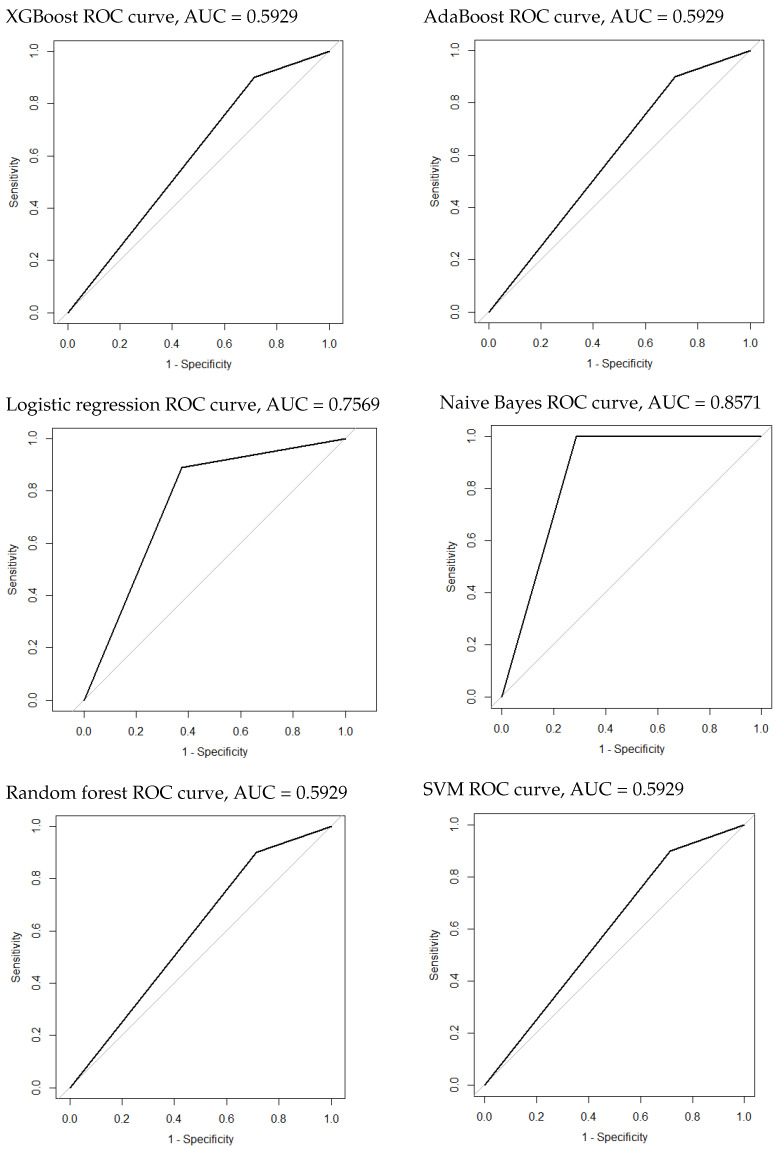
The ROC curves of the classification model of HCC and metastatic tumors.

**Table 1 cancers-16-01158-t001:** Imaging dataset’s details.

Image Size	Slice Thickness	Pixel Spacing	Slice Number
256×256	1.5 mm	0.6172 mm	1200–1700

**Table 2 cancers-16-01158-t002:** Radiomics features in the radiomics analysis.

Types	Features
Shape ( n=14 )	Elongation, Flatness, Least Axis Length, Major Axis Length, Maximum 2D Diameter Column, Maximum 2D Diameter Row, Maximum 2D Diameter Slice, Maximum 3D Diameter, Mesh Volume, Minor Axis Length, Sphericity, Surface Area, Surface Volume Ratio, Voxel Volume
First-order statistics ( n=18 )	10 Percentile, 90th Percentile, Energy, Entropy, Interquartile Range, Kurtosis, Maximum, Mean Absolute Deviation, Mean, Median, Minimum, Range, Robust Mean Absolute Deviation, Root Mean Square, Skewness, Total Energy, Uniformity, Variance
Second-order statistics	
GLCM ( n=24 )	Auto-Correlation, Cluster Prominence, Cluster Shade, Cluster Tendency, Contrast, Correlation, Difference Average, Difference Entropy, Difference Variance, Id, Idm, Idmn, Idn, Imc1, Imc2, Inverse Variance, Joint Average, Joint Energy, Joint Entropy, MCC, Maximum Probability, Sum Average, Sum Entropy, Sum of Squares
GLRLM ( n=16 )	Gray-Level Non-Uniformity, Gray-Level Non-Uniformity Normalized, Gray-Level Variance, High Gray-Level Run Emphasis, Long-Run Emphasis, Long-Run High Gray-Level Emphasis, Long-Run Low Gray-Level Emphasis, Low Gray-Level Run Emphasis, Run Entropy, Run Length Non-Uniformity, Run Length Non-Uniformity Normalized, Run Percentage, Run Variance, Short-Run Emphasis, Short-Run High Gray-Level Emphasis, Short-Run Low Gray-Level Emphasis
GLSZM ( n=16 )	Gray-Level Non-Uniformity, Gray-Level Non-Uniformity Normalized, Gray-Level Variance, High Gray-Level Zone Emphasis, Large Area Emphasis, Large Area High Gray-Level Emphasis, Large Area Low Gray-Level Emphasis, Low Gray-Level Zone Emphasis, Size Zone Non-Uniformity, Size Zone Non-Uniformity Normalized, Small Area Emphasis, Small Area High Gray-Level Emphasis, Small Area Low Gray-Level Emphasis, Zone Entropy, Zone Percentage, Zone Variance
GLDM ( n=14 )	Dependence Entropy, Dependence Non-Uniformity, Dependence Non-Uniformity Normalized, Dependence Variance, Gray-Level Non-Uniformity, Gray-Level Variance, High Gray-Level Emphasis, Large Dependence Emphasis, Large Dependence High Gray-Level Emphasis, Large Dependence Low Gray-Level Emphasis, Low Gray-Level Emphasis, Small Dependence Emphasis, Small Dependence High Gray-Level Emphasis, Small Dependence Low Gray-Level Emphasis
High-order statistics ( n=1584 )	First-Order and Second-Order Features Are Transformed by LoG, Exponential, Square, Square Root, Logarithm, Wavelet (Wavelet-LHL, Wavelet-LHH, Wavelet-HLL, Wavelet-LLH, Wavelet-HLH, Wavelet-HHH, Wavelet-HHL, Wavelet-LLL)

GLCM: gray-level co-occurrence matrix; GLRLM: gray-level run length matrix; GLSZM: gray-level size zone matrix; L; low, H: high.

**Table 3 cancers-16-01158-t003:** Statistical resultsof clinical characteristics between tumor and non-tumor groups.

Clinical Characteristics	Tumor	Non-Tumor	*p*-Value
Sex			0.42
Male	34(36%)	21(22%)	
Female	20(21%)	19(20%)	
Age	55.6±16	51.3±15	0.182

**Table 4 cancers-16-01158-t004:** Model collinearity analysis.

Features	VIF
log.sigma.0.5.mm.3D_firstorder_Maximum	1.226227
log.sigma.2.0.mm.3D_glszm_SizeZoneNonUniformity	4.81969
log.sigma.5.0.mm.3D_glszm_SizeZoneNonUniformity	3.135544
square_glszm_LargeAreaEmphasis	2.184934
logarithm_gldm_DependenceVariance	1.688747
exponential_glcm_ClusterProminence	3.00102
exponential_glcm_ClusterShade	1.598127

**Table 5 cancers-16-01158-t005:** Metrics’ measures for each ML algorithm.

Algorithm	AUROC (95% CI)	Sensitivity	Specificity	NPV	PPV	MCC
SVM	0.8258 (0.677–0.9745)	0.8182	0.8333	0.75	0.8824	0.6419
**NaiveBayes**	**0.9268(0.8224–1)**	0.9091	0.9444	0.9444	0.9091	0.8536
XGBoost	0.8813 (0.75–1)	0.8182	0.9444	0.8947	0.9	0.7785
RF	0.8535 (0.7126–0.9945)	0.8182	0.8889	0.8889	0.8182	0.7071
Logistic	0.8359 (0.6875–0.9842)	0.7273	0.9444	0.8889	0.85	0.7045
AdaBoost	0.8813 (0.75–1)	0.8182	0.9444	0.8947	0.9	0.7785

CI: confidence interval.

**Table 6 cancers-16-01158-t006:** The most important radiomic features in tumor or no tumor classification.

Feature	Importance
log.sigma.2.0.mm.3D_firstorder_Median	0.198357587
wavelet.LLL_glcm_Autocorrelation	0.139106233
log.sigma.2.0.mm.3D_firstorder_Kurtosis	0.066871147
log.sigma.2.0.mm.3D_firstorder_InterquartileRange	0.061976577
logarithm_glcm_ClusterShade	0.055246304
wavelet.LHL_firstorder_Maximum	0.049298670
wavelet.LLH_gldm_SmallDependenceHighGrayLevelEmphasis	0.043135931
exponential_firstorder_MeanAbsoluteDeviation	0.033085765
log.sigma.1.0.mm.3D_glcm_SumAverage	0.029689345
original_glszm_HighGrayLevelZoneEmphasis	0.028640204
log.sigma.1.0.mm.3D_glszm_HighGrayLevelZoneEmphasis	0.024491582
wavelet.LLH_glrlm_LongRunHighGrayLevelEmphasis	0.023100819
wavelet.LLL_firstorder_Kurtosis	0.022962333
wavelet.HHH_glszm_LargeAreaLowGrayLevelEmphasis	0.022767771
log.sigma.2.0.mm.3D_glszm_SizeZoneNonUniformity	0.022381763
log.sigma.2.0.mm.3D_firstorder_MeanAbsoluteDeviation	0.019902419
wavelet.LHL_glcm_ClusterShade	0.018137160
original_firstorder_Mean	0.016972684
log.sigma.0.5.mm.3D_glrlm_LongRunHighGrayLevelEmphasis	0.016380692
log.sigma.5.0.mm.3D_glszm_SmallAreaHighGrayLevelEmphasis	0.013439397
log.sigma.1.0.mm.3D_glszm_GrayLevelNonUniformity	0.013436358
original_glrlm_RunLengthNonUniformity	0.011691586

**Table 7 cancers-16-01158-t007:** Metric measures for each ML algorithm in HCC and metastatic tumor classification.

Algorithm	AUROC (95% CI)	Sensitivity	Specificity	NPV	PPV	MCC
SVM	0.5929 (0.3873–0.7985)	0.2857	0.9	0.6429	0.6667	0.2398
**NaiveBayes**	**0.8571(0.6764–1)**	0.7143	**1**	0.8333	**1**	0.7715
XGBoost	0.5929 (0.3873–0.7985)	0.2857	0.9	0.6429	0.6667	0.2398
RF	0.5929 (0.3873–0.7985)	0.2857	0.9	0.6429	0.6667	0.2398
Logistic	0.7569 (0.5472–0.9667)	0.6250	0.8889	0.7273	0.8333	0.5367
AdaBoost	0.5929 (0.3873–0.7985)	0.2857	0.9	0.6429	0.6667	0.2398

CI: confidence interval.

**Table 8 cancers-16-01158-t008:** The most important radiomic features in HCC and metastatic tumor classification.

Feature	Importance
exponential_glrlm_HighGrayLevelRunEmphasis	0.3296813462
original_shape_Maximum2DDiameterColumn	0.1837066150
original_shape_Maximum2DDiameterRow	0.0666429693
original_shape_Maximum3DDiameter	0.0603642515
exponential_glrlm_GrayLevelVariance	0.0464352022
square_firstorder_RobustMeanAbsoluteDeviation	0.0389244507
exponential_glcm_ClusterTendency	0.0284419206
exponential_glszm_HighGrayLevelZoneEmphasis	0.0253803621
exponential_firstorder_RootMeanSquared	0.0226305816
squareroot_glszm_HighGrayLevelZoneEmphasis	0.0206921639
wavelet.LHL_firstorder_Range	0.0195810157
wavelet.LLH_gldm_GrayLevelNonUniformity	0.0187361912
exponential_glszm_GrayLevelNonUniformity	0.0173371526
original_gldm_GrayLevelNonUniformity	0.0164583230
log.sigma.3.0.mm.3D_glszm_LargeAreaLowGrayLevelEmphasis	0.0151196504
square_glrlm_ShortRunHighGrayLevelEmphasis	0.0127392186
squareroot_firstorder_90Percentile	0.0121420080
logarithm_firstorder_90Percentile	0.0119382485

## Data Availability

The data supporting the conclusions of this article are available upon request from the corresponding author. These data are not publicly accessible due to restrictions on data dissemination imposed by the Ethics Committee.

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
