# Peer review of "Radiomics-Based Classification of Tumor and Healthy Liver on Computed Tomography Images"

_cancers, 2024, doi:10.3390/cancers16061158_

Round 1

Reviewer 1 Report

Comments and Suggestions for Authors

line 31 - What is "plain CT"? This term is ambiguous and not used in the medical literature.

line 33 - "advantageous to young radiologists" - This implies that age rather than experience or confidence in diagnosis impacts radiologist ability to assess liver lesions. I would suggest considering replacing this with "less experienced or less confident" radiologists.

line 36 - this statement is controversial with regards to inducing kidney damage. There is an association with acute kidney injury, primarily in patients with existing chronic kidney disease and only in hospitalized patients. 

line 62 - ROI is not defined. Although a commonly used acronym, this should be defined before using in the text.

line 89 - "three-years experienced radiologists" needs rephrasing.

line 94-94 - consider rephrasing this sentence.

line 99 - radiodensity is one word

line 150 - don't should be changed to "do not"

The study is interesting in that it shows the possibility for radiomics to be applied to liver CT imaging. There are several issues that need to be clarified:

Were all CT scans done with contrast?

Were multiphase CT images examined? What was the pathology of the liver lesions evaluated (HCC, FNH, metastases, cholangiocarcinoma?)

Although the ROC curves show high correlation, are there specific radiomic features which showed higher correlation? 

Comments on the Quality of English Language

There are some sentences that need to be rephrased, with some examples given in the comments. 

Reviewer 2 Report

Comments and Suggestions for Authors

1. In the Introduction: the first sentence with the code reference is not adequate, the following 3 references with same title and published by the same team : a.Lancet.. 2003 Dec 6;362(9399):1907-17. Josep M Llovet Andrew BurroughsJordi Bruix. Hepatocellular carcinoma. Abstract: Hepatocellular carcinoma (HCC) is the fifth most common cause of cancer, and its incidence is increasing worldwide because of the dissemination of hepatitis B and C virus infection.....b.Lancet.2012 Mar 31;379(9822):1245-55. Alejandro Forner 1Josep M LlovetJordi Bruix. Hepatocellular carcinoma. Abstract: Hepatocellular carcinoma is the sixth most prevalent cancer and the third most frequent cause of cancer-related death....c .Lancet.2018 Mar 31;391(10127):1301-1314. Alejandro Forner 1María Reig 2Jordi Bruix 2Hepatocellular carcinoma. Abstract: Hepatocellular carcinoma (HCC) represents approximately 90% of all cases of primary liver cancer, which is the second leading cause of cancer related deaths globally and has an incidence of 850,000 new cases per year.... Which one id better for you?

2. In the 2nd sentence of the Introduction. " Consequently, 41 for tumors suspected to be malignancy, a tissue biopsy is often necessary, despite being 42 invasive, complex, and carrying risks like tumor seeding and bleeding [11]." The concept of the diagnosis of liver malignancy is something different from the current diagnostic guideline. Tissue biopsy for diagnosis of hepatocellular carcinoma is NOT often necessary.

3. Line 48-49, the differential diagnosis for liver tumor, how about the liver adenoma?

4. Line 77-78, " Tumor size Lesions with a maximum diameter 77 of 3 cm or less are left out because they had a low chance of being primary liver tumors" It is not completely right, not low chance but solidly being a primary liver tumor.

5. In the Figure 2. It is better to show the segment of tumor and non-tumor portions

6. It is better to tell us the final goal of this study is to identify the tumor or non-tumor only, or the differentiation of the benign or malignancy? If possible the differenciation between of benign and malignancy is more important in our clinical practice. 

7.  The formation of reference should follow the request from the journal guideline.

Round 2

Reviewer 2 Report

Comments and Suggestions for Authors

1. Please shorten the content of the first paragraph L.19-29.

2.Please make a sentence dissolved into the text intead of listing. L36-39,  and L73-75

3. Please add the information of tumor size and number in your series.

4. Lack the publication year in all References in this manuscript.
